# The Influence of Teacup Shape on the Cognitive Perception of Tea, and the Sustainability Value of the Aesthetic and Practical Design of a Teacup

**Su-Chiu Yang [1,2], Li-Hsun Peng [3,\*] and Li-Chieh Hsu [2]**

[1]   Graduate School of Design, National Yunlin University of Science and Technology, Yunlin 64002, Taiwan; 20180224@fjsmu.edu.cn
[2]   School of Arts and Design, Sanming University, Sanming 365004, China; 20180225@fjsmu.edu.cn
[3]   Department of Creative Design, National Yunlin University of Science and Technology, Yunlin 64002, Taiwan
[\*]   Correspondence: penglh@gemail.yuntech.edu.tw

**Abstract:** The ceramic industry is among the most profitable industries in the world, but, because of the use of nonrenewable materials and high fuel consumption, it also has a carbon footprint. Ceramic materials account for the majority of drinking vessels. Several scholars found that consumers' awareness of drinks and purchasing desires are highly correlated with a vessel's shape and color—in other words, the visual stimulation. However, since prior studies have focused on alcohol, bubble drinks, juice, coffee, cocoa, etc., there has rarely been any research on the appropriate drinking vessels for Chinese tea. This study intends to investigate the visual design of vessels for Chinese tea, in terms of its impact on the taste of the drink, by integrating the thinking and methods of expert users and designers for the sustainability of design and industry. In this study, tea experts and designers were asked for their opinions as a means of data collection. Fuzzy set qualitative comparative analysis (Fs/QCA) was used for data analysis. This study proved that the design of a tea-drinking vessel could have an influence on the perception of the taste and scent of the tea. This research not only brings new meaning to the traditional concept of teacup design, but also reflects famous Japanese craftsman Liu Zongyue's idea of practical beauty, which is beneficial to promoting Chinese tea culture, and contributes to sustainable design and sustainable behavior.

**Keywords:** sustainability; sustainable behavior; design thinking; teacup shape; tea culture; visual recognition; visual design; fuzzy set qualitative comparative analysis

## 1. Introduction

### 1.1. Background Information

Tea, coffee, and cocoa are the three major nonalcoholic drinks in the world, and tea is practically China's national drink. Tea is the most commonly drunk beverage in the world and in 2018, China's tea consumption was approximately 212 million tons, accounting for one-third of the world's tea consumption [1]. Chinese tea culture is deeply rooted in daily life. Over the centuries, the way of drinking tea has also evolved, from boiling tea in the Tang Dynasty, to whisking tea in the Song Dynasty, and finally brewing tea in the Ming Dynasty. As for the tea leaf form, it changed from a tightly packed ball of tea to the contemporary style of loose-leaf tea. Regardless of the tea form or brewing, making a cup of good tea still requires good utensils and water. As the saying goes, "water is the mother of tea and utensils are the father of tea"—a cup of good tea, in addition to good water quality, still relies on the teacups. The teacup not only functions as a carrier of the liquid, but also as a medium of visual aesthetics to improve the quality of the tea-drinking experience. The drinking

vessel for tea has evolved from a tea bowl in the Tang Dynasty, to a teapot in the Song Dynasty, and a teacup in the Ming Dynasty, to the contemporary design of different forms of teacups. As proven over centuries of Chinese tea culture, good tea must find a good match in terms of appropriate utensils. Therefore, in a tea ceremony, the vessel not only serves as a carrier of tea, but also undertakes the function of social interaction, the educational function of aesthetic cultivation, and the integration of added value to the taste and interpretation of tea.

The demand for teasets has increased as a result of tea's continuing popularity. Therefore, relevant issues regarding innovative utensil design have again attracted attention. Thanks to the advancement of science and technology and practical problems arising from rushed design and the production of tea, utensils are often quickly replaced or eliminated. This unnecessary waste of natural resources and the extent of the environmental impact suggest that sustainable design is inadequately understood [2,3]. Issues related to reducing environmental impacts have mostly focused on sustainable design and consumers' sustainable consumption behaviors [4–7]. However, aesthetic and practical design are still challenging for designers, especially in consideration of the sustainability and practical beauty of Chinese teacups.

### 1.2. Motives and Objectives

A well-known Western proverb relating to the food industry, "You eat with your eyes" [8], clearly points to the influence of visual cues on the perception of food. Studies on the impact of visual cues on the taste and the scent of food generally include the color, texture, and eating environment [9–15]. Therefore, when we taste food, in addition to scent and taste, sensory stimulation, such as visual perception, hearing, and touch, all contribute to the construction of a "flavor system" [16] (p. 320) [17]. Many studies have shown that, in addition to the food's texture and appearance, which affect the overall taste and scent, the utensils used in a restaurant or market not only affect visual stimulation, but also have a significant correlation with the appeal of the food [16,18–21]. The difference between drinking a beverage and eating food is the direct contact between the lips and the cup or glass. Therefore, it is worth investigating the relationship between the drinking vessel and the taste it produces. Beverage containers include wine glasses, juice cups, coffee cups, mugs, and teacups. Many scholars have also found a high correlation between consumers' perception of a beverage and the shape and color of the drinking vessel [12,22–27]. However, they mostly focused on alcohol, bubble drinks, juice, coffee, and cocoa. Spence and Wan [28] also pointed out that, in the tea-drinking population around the world, few researchers have really studied the appropriateness of vessels for tea; specifically, there is a lack of in-depth research into Chinese teacups, which is basically the motivation for this study.

Chinese tea culture has a long history and it is not difficult to notice the matching of teacup and tea leaves and the fact that the shape of a teacup can change due to the appearance of the tea leaves and the way of making tea. Despite the wide variety of teacup designs on the market from ancient times to the contemporary period, most consumers, tea merchants, and teachers of the tea ceremony place the emphasis on the way of making tea and the selection of appropriate ceramic utensils [29–31], or on the tea culture and innovative form, as focused on by teacup designers or ceramic artists; few are willing to investigate the relationship between teacup design and the quality of the tea.

In terms of product materials, ceramics are more environmentally friendly than plastic. However, the emergence of new teaware designs has also triggered discussions among many scholars on negative environmental issues, such as production waste, consumerism, and energy waste [32–35]. Data showed that the ceramic industry is among the most profitable industries in the world, but it also has a high carbon footprint due to its use of raw nonrenewable materials and high fuel consumption [36]. Therefore, these conventional industries focus on designing products that require the least energy to be produced and products that can be recycled [37]. However, "at the current time, material consumption of natural resources is increasing, particularly within Eastern Asia" [38]. Currently, many studies emphasize sustainable design's capacity to change user behavior and attain environmental benefits through the interaction of product and user behavior [39–41].

From the perspective of product concept design, the thinking and methods of designers and engineers are different, focusing on visual attributes and performance, respectively [42]. Yu et al. [43] also found that cross-disciplinary collaboration in design at the stage of new product development (NPD) led to better performance. Therefore, from the viewpoint of environmental and business sustainability, the author believes that, in the tea-drinking process, the interaction between teacup design and the taste of the tea should be explored to help with an understanding of how to design practical teacups in line with Liu Zongyue's ideology of practical aesthetics as well as provide an opportunity for tea drinkers to contribute to environmental sustainability through their collective behaviors. On the other hand, "innovation and creativity are high on the agenda of society in general and design and business in particular" [44] (p. 27). Considering successful and sustainable businesses, an innovative and creative design is absolutely vital [45]. This study focuses on the relationship between user behavior, user experience, and products through a user-centered design and interaction [46]. Therefore, the possible influence of teacup selection on the quality of tea and the tea-drinking experience is the major focus of this research. Specifically, this study investigates:

1. The impact of utensil design on the intensity of tea taste.
2. The impact of utensil design on the bitterness of tea.
3. The impact of utensil design on the astringency of tea.
4. The impact of utensil design on the sweet aftertaste of tea.
5. The impact of utensil preference on the overall presentation of tea.

*1.3. Research Subjects*

1.3.1. The Way of Tea Drinking, and Tea-Drinking Utensils

Chinese tea culture stretches back over 5000 years, but it was not until the book *The Classic of Tea Lu, Yu* was published during the Tang Dynasty that the art of tea-drinking and the overall tea culture started to be standardized. In terms of vessels during the Tang Dynasty, Lu Yu personally favored a celadon tea bowl. As for the whisking technique in the Song Dynasty, the vessels were mostly black glazed tea bowls with a mouth diameter of 12 to 13 mm [47]. During the Ming Dynasty, the change from brick tea to loose-leaf tea was a milestone in Chinese tea culture, which further influenced the modern way of making tea. Pure white and small teacups were better for tea drinking [48]. Furthermore, different teacups were used in different seasons [49], so that the size and aesthetics of teaware was not just a functional decision but also a cultural and artistic one. The modern tea-drinking practice is heavily influenced by Western culture, becoming more diversified in terms of the method and teacup design. English afternoon tea, the Japanese tea ceremony, innovative beverages, and flower or fruit teas are part of this diversity. This study focused on the traditional method of placing an appropriate amount of tea leaves in a teapot for brewing before pouring it into smaller teacups for observation of the color, smelling and tasting it.

1.3.2. Sensory Evaluation of Tea

The quality of tea can be objectively determined by a sensory evaluation of the liquid. The evaluation items include the overall appearance, color, aroma, taste, and the used leaves. This is also the order of evaluation: first, observe the appearance and the color of the liquid; then, smell and sip the tea for the taste; lastly, examine the brewed leaves. The taste and texture of the tea can be classified into 12 categories: strong, fresh, refreshing, sweet, smooth, rich, mellow, dull, coarse but plain, coarse but astringent, immature but astringent, bitter and astringent, and plain like water [50]. Generally, a tea drinker will simply observe the texture directly, smell the aroma, and savor the taste of the tea. However, with the 12 evaluation items, it is possible to scientifically determine the quality of the tea. The standard evaluating procedure involves placing 5 g of tea leaves in a 150-cc white porcelain cup topped up with 100 °C water for 5 min. Then, the tea is poured out for inspection and assessment.

The taste buds on the tongue are the most sensitive tool for identifying the taste of tea, and transfer the response of stimulus to the central nervous system to produce a sensation of taste. Excessive temperature will damage the taste buds' ability to complete the evaluation. The most appropriate temperature for tea is 40–50 °C; thus, it is better to observe the appearance and color of the tea first, along with its scent, before proceeding to sip the tea. The Chinese character "taste" (i.e., savor) consists of the character for "mouth" repeated three times, suggesting that, generally, a consumer should enjoy the tea in three separate sips of 5 cc each. Thus, the best teacup size is around 15 cc [51].

## 2. Literature Review

### 2.1. Impact of Visual Cues on Taste

The human eyes, ears, nose, tongue, and body are responsible for the five senses of sight, hearing, smell, taste, and touch. According to the organ structure and the complexity of their functions in terms of their distance to acquiring sensory information, the visual and auditory senses are referred to as "advanced" or simply "distance" of sensation, while taste is often regarded as a "lower grade" of sensation, or called "contact (close) perception" [52]. In daily life, the brain will accept sensory stimulation information and repetitive stimulation will gradually form memories through a bottom-up process. Such cognitive awareness of similar stimulation in the future will activate the memory through a top-down process [53,54]. In other words, it is a result of activating cognitive awareness from external stimulation and represents the information processing in the direction of brain to the outside world, which is thus referred to as a top-down process. Moreover, through the body's sensory receptors, the generated sensation will pass through the top-down process to become a cognitive perception in the brain. Then, the cognitive perception is reflected as different sensations to the receptor. The senses of sight, hearing, smell, taste, and touch are separate, but have an interactive influence on each other (Figure 1). A "cognitive hierarchy" represents the information processing of complex psychological activities to accept, focus, recognize, memorize, understand, and reflect upon the stimulus from the multisensory receptors [55–58]. Through associative learning and the development of synesthesia [52], these actions will reinforce the product experience [59].

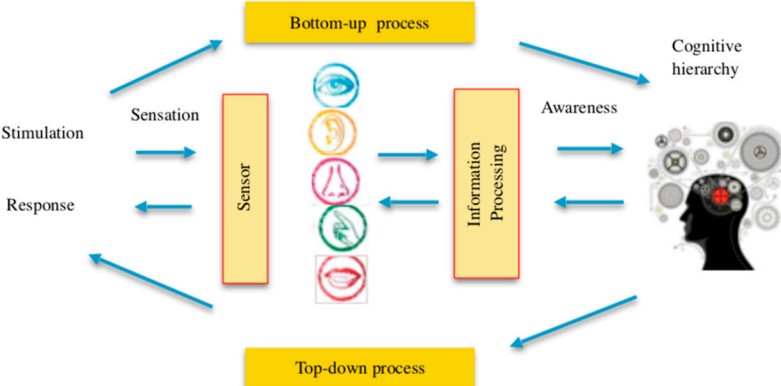

**Figure 1.** Stimulus response and the flow chart of sensory perception (plotted for this study).

Product experience is the perception of the psychological effect of interaction with the product, including the extent to which all senses are stimulated, the meaning and value we attach to the product, and the feelings and emotions that are triggered. Specifically, the product experience is not necessarily limited to an actual physical activity, but may involve passive visual perception, or even memory [60]. People's reaction to a taste is typically a complex combination of perceptions of smell, sight, taste, and touch through the process of product experience [9]. However, in terms of assessing the external environment, obtaining external information or knowledge, and consumer motivation, most people are visually oriented. According to statistics, visual cues account for 87% of sensory information; thus,

visual cues are believed to be the most important to the five senses [52,53,61–63]. Therefore, in addition to the presentation of the food and beverage itself, visual cues play an important role in affecting the taste perception, which is in line with the saying, "You eat with your eyes." Another influential factor is the tactile sensation when in contact with the vessel, because human "tactile receptors are mostly distributed on the nose, lips, and fingertips" [53], and these body parts are in constant, direct contact with the utensil during eating and drinking. Therefore, Piqueras-Fiszman and Spence [12] proposed the concept of "sensation transference" to explain how the color, texture, weight, and shape of a vessel affect consumers' perception of the beverage inside. Related studies have shown that the material of the vessel could affect the subject's experience of the drink. For example, Schifferstein [64] found that participants' perception of drinking hot tea was biased towards the use of ceramic materials, which coincided with the practical experience of using such materials in the Chinese tradition of tea-drinking.

## 2.2. The Visual Significance of Chinese Teacups

The semiotics of Ferdinand de Saussure classifies objects into forms and concepts. The physical form of an object is called a signifier, and the psychological abstraction of an object is called the signified. The relationship between the two is arbitrary, and it is customarily defined in social or cultural conventions. The form and concept of an object can be transformed and linked in meaning through "associative relations" and "syntagmatic relations." The contemporary teacup in Chinese tea culture has evolved differently from the Western mug. The shape and size may vary according to the type of tea and the time period. Thus, as a signifier, a teacup is a direct reflection of its purpose of holding tea, but diverse designs, as reflections of different artistic meanings, with the addition of an audience's perception, will transform the relationship to have different connotations that may vary greatly between people with different interpretations [65]. As seen in historical books on tea culture (the relationship between teacup color/design and the texture of the liquid is described in the *Treatise on Tea*, the appearance of the tea is related to the teacup color and size in the *Tea Shu*, and the meaning of a teacup's artistic design is discussed in the *Book of Chaozhou Tea*), Chinese teacups have undergone a symbolic evolution in terms of appearance and design.

Therefore, a teacup functions as a signifier for drinking tea and is visually signified by its design and color; in the process of visual cognition, the transference of sensations of color and shape will occur to influence perceptions of the color, aroma, and taste of the tea [66].

## 2.3. Outer Appearance and Design of Tea Vessels

Multiple studies showed [12,26,67] that the design of the vessel affected the judgment of a beverage's color, as well as the expectations for its taste and aroma. For example, a purple drink in a wine glass would be considered wine, while a purple drink in a cylindrical glass would be considered grape juice. The procedures for tea tasting and wine tasting are the same, and involve smelling the aroma before tasting. Therefore, previous studies have tried to test the effect of seeing but not touching the wine cup on the recognition of aroma. When subjects could not see or touch the wine cup, they were unable to recognize the aroma. However, when the subjects saw and held the wine cup, its design had a certain degree of influence on their recognition of the wine aroma [68–70]. In addition, some studies found that even experienced wine drinkers were still influenced by the shape of the wine glass [71,72]. Furthermore, even professional tea ceremony teachers graded differently the aroma and taste of tea made from the same tea leaves but placed in teacups of different designs. Besides wine, Doorn et al. [23] conducted cross-cultural research on coffee cups in China, Colombia, and the United Kingdom, analyzing factors affecting the coffee's taste, and found that visual information such as the size and height of the coffee cup was quite relevant to consumers' expectations.

## 2.4. Color of Vessel

Studies have found that the color of the container affected consumers' judgment of the taste and quality of the food or drink [13,18,60]. As for the influence on beverages, Van Doorn, Wuillemin, and

Spence [25] showed that the appearance and color of the cup would alter subjects' sensory perception of the same concentration of coffee. The results showed that the same coffee in a white ceramic mug would look more intense in taste than in a transparent mug. In addition to the perception of taste, the color of the cup also affects the perception of coffee temperature [73] and the perception of water [74]. Piqueras-Fiszman and Spence [12] have shown that the color of the vessel used for food and beverages enhances the flavor and aroma. The study found that subjects drinking the same hot cocoa would perceive a difference in the sweet aftertaste based on the color of the cup. However, the way of drinking Chinese tea is different from that of coffee, cocoa, wine, and other beverages. No matter whether it is a professional tea evaluation expert or a common consumer, there is an established order of appreciation and evaluation: first observing the appearance and color of the tea, before proceeding to smell it and sip it for the taste. Hence, in order not to affect the judgement of the color of the tea, this study controlled for the color variable and used white porcelain teacups.

*2.5. Impact of Tactile Cues on Taste*

During eating, in addition to the impact of visual cues on taste, tactile receptors are the most densely populated sensors on hands and lips. Research had indicated that haptic interaction created increasing immersion and involvement with tangible objects [75,76]. When these tactile receptors touch the utensil to arouse sensation, an association begins with memories stored in the brain to produce a psychological response that affects the taste [51,52,77]. Liu Zongyue, the father of Japanese folk art research, once said: "The beauty of craftsmanship is the beauty of practicality, which is the combination of service and beauty." By holding the teacup and bringing it into contact with the lips, the warmth of the tea and the vessel will be conveyed to the user. Therefore, in terms of teacup design, besides its visual perception, it is also necessary to explore the impact of the tactile sensation of holding the cup and touching it to the lips on the taste of the tea. The thickness of the teacup, the thickness of the rim, and even the size and tactile sensation of the teacup on the hands and lips can all affect the psychological response to the taste of the beverage inside.

In view of the deeply rooted culture of tea-drinking and the advancements in planting and manufacturing tea in recent years in Taiwan, tea culture has once again captured the attention of the public, especially due to the contribution of design to tea functions. In the design of a tea function, the selection of utensils will not only influence visual aesthetics, but more importantly the taste. In addition to the teapot, tea drinkers will most likely be in direct contact with the teacup, which thus serves as both a tactile and visual carrier of the taste and aroma of the tea. Therefore, the teacup design and the tactile sensation on hands and lips were the factors selected for this study.

## 3. Research Method

*3.1. Research Framework*

The purpose of this study was to explore the influence of Chinese tea utensils and teacup design on the taste and aroma of tea by attempting to discover a causal relationship through experimentation. The independent variable was the shape of the teacup, while the dependent variables included the sweet aftertaste and overall presentation of the tea. Through a literature review on historical utensils and appropriate teacup design, a questionnaire was formulated according to the classification standards set by the Tea Research and Extension Office of Agriculture and Food Agency of Executive Yuan. Conducting a Fs/QCA (fuzzy set qualitative comparative analysis) on the experimental results helped us to quantify the theoretical shortcomings of using a single method, and issues such as the loss of focus when analyzing qualitative data. After the tests, interviews were conducted and the results cross-analyzed with the experimental data. The steps of this study are shown in Figure 2 and the results are described and discussed in Section 4.

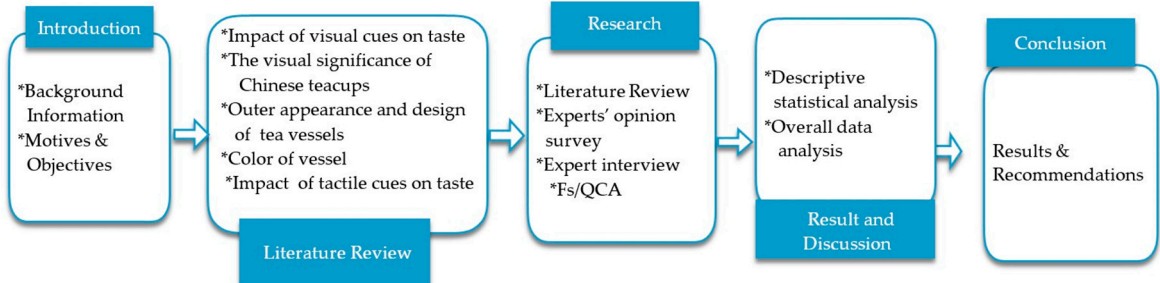

**Figure 2.** Research schematic.

### 3.2. Experiment Design

#### 3.2.1. Subjects

As specified by the Tea Research and Extension Office of Agriculture and Food Agency of Executive Yuan, the taste and texture of tea can be classified into 12 categories: strong, fresh, fresh, refreshing, sweet, smooth, rich, mellow, dull, coarse but plain, coarse but astringent, immature but astringent, bitter and astringent, and plain like water. Such a professional classification is typically not recognized by general consumers. To achieve the desired accuracy and effectiveness, the study used the fuzzy set qualitative comparative analysis, which would not be restricted by the small sample size. The optimal sample size was 10 to 12. In this study, considering the user-centered design, the participants were subjectively selected for their professional expertise or skills related to tea or ceramics design. There were 12 experts interviewed, introduced by sex, age, and professional background in Table 1. They included nine females and three males; there are five tea ceremony teachers, one merchant, three tea critics and professional teachers of the tea ceremony, one tea maker and tea critic, one professional teacher of the tea ceremony and ceramic designer, and one tea merchant and professional teacher of the tea ceremony. The male professional teacher of the tea ceremony and ceramic designer also owns a ceramic factory. The tea ceremony teachers and tea critics were certified, as well as the two senior merchants and the tea maker.

**Table 1.** Basic information on subjects.

| Number | Professional Background | Sex | Age |
|--------|-------------------------|-----|-----|
| 1 | Professional teacher of tea ceremony and ceramic designer | Male | 50 |
| 2 | Professional teacher of tea ceremony | Female | 37 |
| 3 | Professional teacher of tea ceremony | Female | 49 |
| 4 | Professional teacher of tea ceremony | Female | 67 |
| 5 | Professional teacher of tea ceremony | Female | 47 |
| 6 | Professional teacher of tea ceremony | Female | 60 |
| 7 | Tea critic and Professional teacher of tea ceremony | Female | 55 |
| 8 | Tea critic and Professional teacher of tea ceremony | Female | 42 |
| 9 | Tea critic and Professional teacher of tea ceremony | Male | 57 |
| 10 | Tea merchant | Female | 40 |
| 11 | Tea merchant and Professional teacher of tea ceremony | Female | 42 |
| 12 | Tea maker and Tea critic | Male | 59 |

#### 3.2.2. Experimental Materials

Tea can be fully fermented, partially fermented, or totally nonfermented. In this study, partially fermented tea (accounting for the majority of manufactured tea) was used and Lishan Oolong, one of the well-known Taiwanese teas, was chosen as the material, as its taste and scent could be easily distinguished. The way of making tea followed the regulations set forth by the Tea Research and Extension Office of Agriculture and Food Agency of Executive Yuan: 3 g of tea were fully immersed

in 100 °C water in a 150-cc white porcelain cup for 5 min. The resulting tea was used as the experimental material.

### 3.2.3. Objectives

The primary objective of the study was to investigate if the design of the primary tea utensil, the cup, would have any impact on the bitterness, sweet aftertaste, and astringency of the tea. The teacup design could trigger visual and oral tactile sensations. Thus, both the visual and the tactile sensations were included. Although in previous studies the color of the teacup was also a visual variable affecting the taste, the color of the tea is the most important factor while drinking Chinese tea. Therefore, to prevent the influence of the vessel color, a white porcelain cup was used as a control so that the color of the tea would not vary across subjects. The selection criteria for teacup design are explained below.

#### Visual

In *Chaozhou Tea Classic*, the teacup design could vary by season, such as a "bull's-eye" cup in spring, a chestnut cup in summer, a lotus cup in autumn, and a top-down bell-shaped cup in winter (as seen in Figure 3). Therefore, the cup design was customized to the different seasons based on associative learning of sensory stimulation, recognition, and memory. The *Far-Reaching Fragrance of Tea*, published by the National Palace Museum, analyzes and classifies teacup design based on the mouth and the interior cup belly, as explained in detail below [78].

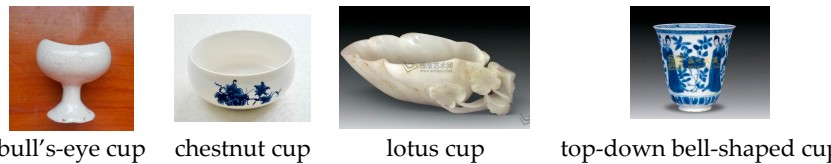

bull's-eye cup  chestnut cup  lotus cup  top-down bell-shaped cup

**Figure 3.** The shape of the cup rim (pictures taken from https://wenda.so.com/q/1470565348724431).

1. The mouth

The mouth of a teacup can be classified as contracted, narrow, open, or flaring (as seen in Figure 4). A contracted mouth has the rim converging inward. A narrow mouth has a rim skirt around the opening. An open mouth has an outward extension of the rim and is slightly bent at the lips. The design of a teacup can be a "bull's-eye" cup, a chestnut cup, a lotus cup, or a top-down bell-shaped cup.

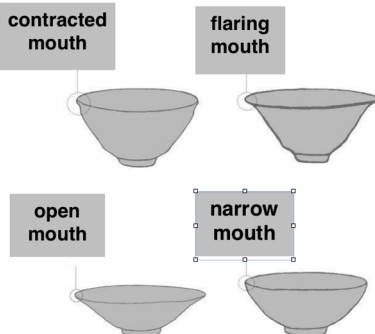

**Figure 4.** The shape of the cup rim.

2. Interior cup belly

The interior cup belly can be classified as deep or shallow, which is often a reflection of a teacup's height. A wide or narrow belly refers to the size of the opening of the teacup.

Tactile

Hands and lips are in direct contact with the teacup to produce a tactile sensation. Therefore, the texture, weight, and thickness of the cup are also factors that influence the choice of cup. According to the comprehensive visual and tactile factors, the teacups selected in this study included: A cup: flaring mouth, wide belly (opening width: 8.2 cm, height: 3 cm); B cup: contracted mouth, shallow belly (opening width: 6 cm, height: 3.5 cm); C cup: narrow mouth, shallow belly (opening width: 5 cm, height: 2 cm); D cup: open mouth, narrow belly (opening width: 6.1 cm, height: 2.8 cm); and E cup: open mouth, deep belly (opening width: 4.5 cm, height: 3.9 cm) (Figure 5).

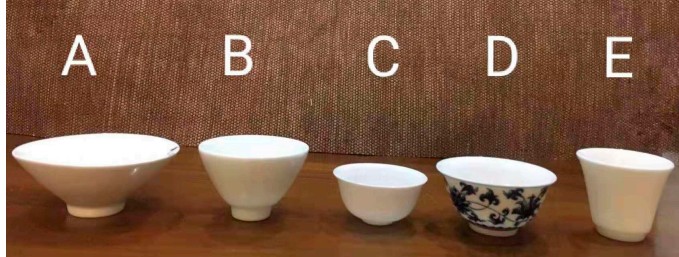

**Figure 5.** Cups A to E (pictures taken by the authors).

### 3.2.4. Questionnaire

As specified by the Tea Research and Extension Office of Agriculture and Food Agency of Executive Yuan [50], the taste and texture of tea can be classified into 12 categories: strong, fresh, fresh, refreshing, sweet, smooth, rich, mellow, dull, coarse but plain, coarse but astringent, immature but astringent, bitter and astringent, and plain like water. The survey instead classifies them as: (1) strength of taste; (2) intensity of bitterness; (3) intensity of astringency; (4) intensity of sweet aftertaste; (5) overall tea presentation; and (6) degree of preference of teacup, in accordance with a 5-point Likert scale.

### 3.2.5. Experimental Environment

The Tea Research and Extension Office of Agriculture and Food Agency of Executive Yuan specified the environment, with natural lighting as the best testing setting. There should be adequate and uniform lighting, but not directly overhead. A room in the northern hemisphere should face north to have the best natural light. For an environment without adequate lighting, fluorescent lighting can be added as a supplement on the table. To allow subjects to easily observe the color of the tea, the selected room would face north, with the table placed next to a window for natural lighting. To prevent cooling of the tea, which would interfere with the judgment of scent and taste, a fanless air conditioner was used to maintain the room at 22 to 24 °C. The room was kept clean without any odors.

### 3.2.6. Experimental Operation

Subjects in this study were informed in advance to wash their mouth out to prevent any residual tastes. They were also informed of the entire experimental procedure and the content of the questionnaire. Afterwards, the materials described in Section 3.2.2 were placed in teacups and prepared with hot water before being delivered to the subjects, who would drink the teas in order and then make subjective judgments on the taste of the tea and the design of the teacups. These results were also recorded in the survey. Two subjects were tested in each session and given enough time to fill out the questionnaire. To maintain the objectivity of drinking order and the results, the order of the teacups differed for each subject.

*3.3. Fuzzy Set/Qualitative Comparative Analysis*

Unlike most quantitative studies, which are based on causality, the fuzzy set/qualitative comparative analysis (Fs/QCA), developed by the social scientist Charles Ragin, is a method of obtaining linguistic summaries from relational data between cases. The basic concepts differ from standard statistical analysis techniques, relying on correlations to determine causality and significance tests to assess generalizability. Fs/QCA seeks to establish a logical link between causal conditional combinations (common causality) and outcomes, with the result of summarizing the causality of all possible combinations of subsets of causal conditions (or their complements), and the results [79]. Ragin and Sedziaka [80] showed that Fs/QCA uses combinatorial logic, based on fuzzy set theory and Boolean logic minimization, to produce combinations of results that may be sufficient or necessary. Fs/QCA is best applied when the researcher assumes that complex causality is present but the number of cases is too low for statistical techniques. Therefore, this study analyzed the influence of the teacup on the taste, bitterness, sweet aftertaste, astringency, and overall presentation of the tea, as a way of compensating for the traditional statistical method that only reveals correlations among variables but ignores other weak links. Fs/QCA can point out the relationship of sufficient but not necessary conditions, and is not limited to small sample sizes. Zhang [81] mentioned that for expert interviews in Fs/QCA, the minimum number of participants should be 10 or more. In this study, there was a variety of visual elements to the teacup design and a complex mix of taste, bitterness, sweet aftertaste, astringency, and overall presentation of tea was observed, which common consumers might have difficulty in interpreting. Considering the complex causality and multiple conditions of the visual perception factors, expert experimental subjects were chosen to gather data. Anttila [82] had pointed out that empirical data in a qualitative research can be used as a basis for a description of reality on a complex, holistic, and systemic vision. Therefore, it was suitable to use a multilevel qualitative narrative of Fs/QCA to establish a truth table of causal relationship to explore the relational link. The interpretation parameters of the Fs/QCA analysis results included: (1) Solution consistency, which was equivalent to the significance of quantitative analysis, and must be greater than 0.75. The explanatory power was stronger when approaching 1 to judge the degree of causality of the conditional combination. (2) Solution coverage, which could explain the explanatory power of causality, but without a specific standard. Generally, the consistency value is three times the coverage value. Therefore, this study used Fs/QCA software to generate a truth table for the causal relationship between teacup shape and various sensations of taste, bitterness, sweet aftertaste, and astringency, in order to determine the correlation between consistency and coverage.

## 4. Results and Discussion

*4.1. Fs/QCA Statistical Analysis*

4.1.1. Descriptive Statistical Analysis

A total of 12 experts (nine females and three males: five tea ceremony teachers, one merchant, three tea critics and professional tea ceremony teachers, one tea maker and tea critic, one professional tea ceremony teacher and ceramic designer, and one tea merchant and professional teacher of tea ceremony) answered the questionnaire about the taste, astringency, bitterness, sweet aftertaste, and overall presentation of the tea, as well as their preference among the five teacups. After the questionnaire, a personal interview was conducted.

The five teacups were as follows: A cup: flaring mouth, wide belly; B cup: contracted mouth, shallow belly; C cup: narrow mouth, shallow belly; D cup: open mouth, shallow belly; E cup: open mouth, deep belly. Based on the descriptive statistical analysis (Tables 2–7) of the above variables, it was found that the E cup (open mouth, deep belly) had the highest average for taste, intensity, sweet aftertaste, and preference of teacup, scoring 4.333333, 4, 3.916667, and 4, respectively. This meant that E was favored by most subjects as offering the strongest taste, bitterness, astringency, and sweet

aftertaste. The average scores of the A cup (flaring mouth, wide belly), in terms of taste, bitterness, astringency, and sweet aftertaste, were lower than 3, which meant that most subjects thought the taste to be weak, not bitter, not astringent, and without a sweet aftertaste, and the overall presentation of the tea in this vessel was only average. The average scores of the B cup (contracted mouth, shallow belly), in terms of taste, bitterness, astringency, and sweet aftertaste, were lower than 3, suggesting that most subjects thought the taste was weak, not bitter, and not astringent, with a fairly average sweet aftertaste and overall presentation. The average scores of the C cup (narrow mouth, shallow belly), in terms of astringency and sweet aftertaste, were lower than 3, so most subjects felt the tea was not astringent and had no sweet aftertaste, even though it conveyed fairly average taste and bitterness. The D cup (open mouth, narrow belly) had the highest average in terms of the intensity of astringency. This suggests that most subjects felt a sensation of astringency when drinking tea from this cup. The standard deviation of each variable in Tables 2–7 indicates the degree of variation, where a larger value suggests more divergence. The standard deviations of this table were between 0.7071068 and 1.621141, which suggests that subjects felt differently in terms of the taste, bitterness, astringency, sweet aftertaste, overall presentation, and teacup preference. This also implies that tea preferences are subjective.

**Table 2.** Descriptive statistical analysis of intensity of tea taste.

| Variable (Teacup) | Mean | Std. DEV | Minimum | Maximum | *N* Cases |
|---|---|---|---|---|---|
| A | 2.83333 | 1.404358 | 1 | 5 | 12 |
| B | 2.666667 | 0.8498366 | 1 | 4 | 12 |
| C | 3.25 | 0.9359664 | 1 | 4 | 12 |
| D | 3.583333 | 1.320248 | 1 | 5 | 12 |
| E | 4.333333 | 1.312335 | 1 | 5 | 12 |

**Table 3.** Descriptive statistical analysis of intensity of tea bitterness.

| Variable (Teacup) | Mean | Std. DEV | Minimum | Maximum | *N* Cases |
|---|---|---|---|---|---|
| A | 2.083333 | 1.114924 | 1 | 5 | 12 |
| B | 2 | 0.9128709 | 1 | 4 | 12 |
| C | 3.416667 | 1.114924 | 1 | 5 | 12 |
| D | 3.666667 | 0.942809 | 1 | 5 | 12 |
| E | 4 | 1.080123 | 1 | 5 | 12 |

**Table 4.** Descriptive statistical analysis of intensity of astringency of tea.

| Variable (Teacup) | Mean | Std. DEV | Minimum | Maximum | *N* Cases |
|---|---|---|---|---|---|
| A | 2.5 | 1.190238 | 1 | 5 | 12 |
| B | 2.333333 | 1.027402 | 1 | 4 | 12 |
| C | 2.75 | 1.163687 | 1 | 5 | 12 |
| D | 4 | 0.7071068 | 1 | 5 | 12 |
| E | 3.916667 | 1.114924 | 1 | 5 | 12 |

**Table 5.** Descriptive statistical analysis of intensity of sweet aftertaste of tea.

| Variable (Teacup) | Mean | Std. DEV | Minimum | Maximum | *N* Cases |
|---|---|---|---|---|---|
| A | 2.666667 | 1.312335 | 1 | 5 | 12 |
| B | 3.166667 | 1.067187 | 1 | 5 | 12 |
| C | 2.666667 | 0.8498366 | 1 | 4 | 12 |
| D | 2.916667 | 1.187317 | 1 | 5 | 12 |
| E | 3.916667 | 1.320248 | 1 | 5 | 12 |

**Table 6.** Descriptive statistical analysis of overall presentation of tea.

| Variable (Teacup) | Mean | Std. DEV | Minimum | Maximum | *N* Cases |
|---|---|---|---|---|---|
| A | 3.333333 | 1.490712 | 1 | 5 | 12 |
| B | 3.416667 | 0.7592028 | 2 | 5 | 12 |
| C | 3.333333 | 1.178511 | 1 | 5 | 12 |
| D | 3.333333 | 1.312335 | 1 | 5 | 12 |
| E | 3.166667 | 1.621141 | 1 | 5 | 12 |

**Table 7.** Descriptive statistical analysis of teacup preference.

| Variable (Teacup) | Mean | Std. DEV | Minimum | Maximum | *N* Cases |
|---|---|---|---|---|---|
| A | 3.666667 | 1.027402 | 2 | 5 | 12 |
| B | 3.5 | 1.118034 | 1 | 5 | 12 |
| C | 3.583333 | 0.9537936 | 1 | 5 | 12 |
| D | 3.583333 | 0.9537936 | 2 | 5 | 12 |
| E | 4 | 1.080123 | 1 | 5 | 12 |

### 4.1.2. Comprehensive Data Analysis

The use of Fs/QCA software presented the causal relationships of the taste, bitterness, sweet aftertaste, astringency, and overall presentation of tea with five different teacup designs. The dependent variables included the taste, bitterness, sweet aftertaste, astringency, and overall presentation of the tea. The independent variable was the teacup preference. The consistency and coverage were analyzed as illustrated in Table 8. According to the interpretation parameters of the Fs/QCA analysis results designed by Ragin [82], a consistency greater than 0.75 would have more explanatory power, suggesting a significant causal relationship in the conditional combination. Table 7 shows that the consistency was higher than 0.75; the lowest was 0.818, and the highest was 1, indicating that the teacup design preference was consistently related to the taste, bitterness, sweet aftertaste, astringency, and overall presentation of the tea. The explanatory power of solution coverage for causality is generally one-third of the value of consistency, which indicates the power to interpret a causal relationship. Table 8 shows the minimum coverage at 0.425 and the maximum coverage at 0.826, indicating that the teacup design preference could account for the taste, bitterness, sweet aftertaste, astringency, and overall presentation of the tea. Therefore, the subject's preference of teacup shape was sufficient to explain the impact on perception of the taste, bitterness, sweet aftertaste, astringency, and overall presentation of the tea. At this first stage of data analysis, our results are in line with the conclusion in Yu el al. [43] that user experience is a major concern for designers.

**Table 8.** Descriptive statistical analysis of the taste, bitterness, astringency, sweet aftertaste, and overall presentation of tea.

| | A Consistency/ Coverage | B Consistency/ Coverage | C Consistency/ Coverage | D Consistency/ Coverage | E Consistency/ Coverage |
|---|---|---|---|---|---|
| | | | Taste of tea | | |
| Preference of teacup | 0.863/0.593 | 1.00/0.724 | 0.952/0.571 | 0.954/0.552 | 0.952/0.425 |
| | | | Bitterness of tea | | |
| Preference of teacup | 0.818/0.782 | 0.904/0.826 | 0.952/0.555 | 0.954/0.525 | 0.952/0.465 |
| | | | Astringency of tea | | |
| Preference of teacup | 0.863/0.703 | 0.904/0.730 | 0.952/0.645 | 1.000/0.500 | 0.952/0.425 |
| | | | Sweet aftertaste of tea | | |
| Preference of teacup | 0.909/0.645 | 1.000/0.600 | 0.952/0.689 | 0.909/0.645 | 0.952/0.476 |
| | | | Overall presentation of tea | | |
| Preference of teacup | 0.863/0.542 | 1.000/0.552 | 0.952/0.555 | 0.909/0.571 | 0.904/0.558 |

In order to establish a causal connection between the shape of the teacup and the taste, bitterness, sweet aftertaste, astringency, and overall presentation of the tea, a truth table of causality was constructed that combined all possible criteria under the causal condition. This study defined the overall presentation of the tea as the dependent variable, while the teacup's visual design, along with the taste, bitterness, sweet aftertaste, and astringency were all independent variables used to explore the causality. Table 9 shows that the combination of a teacup's visual design with the taste, bitterness, sweet aftertaste, and astringency was consistently related to the overall presentation of the tea, as the minimum consistency was 0.882 and the maximum was 1, both values exceeding the minimum of 0.75 designated by Ragin [83]. This indicated the significant causality of these factors. The coverage value also suggested a significantly positive and consistent relationship. In other words, in this study, using five different teacup designs (A to E), as analyzed by experts' experimentation and Fs/QCA, we found that the overall presentation of the tea was significantly influenced by the visual design of the teacup and the taste, bitterness, sweet aftertaste, and astringency of the tea, while each factor alone was adequate to affect the overall presentation of the tea. Hence, the preference of teacup design could affect the taste, bitterness, sweet aftertaste, astringency, and overall presentation of the tea, proving Piqueras-Fiszman and Spence's [12] proposal of "sensation transference" for explaining the sensory characteristics of the color, texture, weight, and shape of a beverage vessel, which would supposedly affect consumers' perception of the contents.

**Table 9.** Descriptive statistical analysis of the overall presentation of tea.

| | Overall Presentation of the Tea (Outcome) | | | | |
|---|---|---|---|---|---|
| Analysis Item | A Consistency/ Coverage | B Consistency/ Coverage | C Consistency/ Coverage | D Consistency/ Coverage | E Consistency/ Coverage |
| Taste/Bitterness/ Astringency/Sweet aftertaste/Teacup preference | 0.866/0.371 | 1.000/0.447 | 1.000/0.500 | 0.947/0.514 | 0.947/0.529 |
| Bitterness/Astringency/ Sweet aftertaste/Teacup preference | 0.875/0.400 | 1.000/0.447 | 1.000/0.500 | 0.947/0.514 | 0.947/0.529 |
| Taste/Bitterness/Sweet aftertaste/Teacup preference | 0.875/0.400 | 1.000/0.500 | 1.000/0.528 | 0.947/0.514 | 0.947/0.529 |
| Taste/Bitterness/Astringency/ Sweet aftertaste | 0.866/0.371 | 1.000/0.447 | 1.000/0.528 | 0.947/0.514 | 0.900/0.529 |
| Taste/Bitterness/Astringency/ Teacup preference | 0.875/0.400 | 1.000/0.447 | 1.000/0.500 | 0.95/0.542 | 0.947/0.529 |
| Taste/Astringency/Sweet aftertaste/Teacup preference | 0.866/0.371 | 1.000/0.500 | 1.000/0.528 | 0.95/0.542 | 0.947/0.529 |
| Taste/Bitterness/Sweet aftertaste | 0.875/0.400 | 1.000/0.500 | 1.000/0.556 | 0.947/0.514 | 0.900/0.529 |
| Taste/Astringency/Sweet aftertaste | 0.866/0.371 | 1.000/0.500 | 1.000/0.556 | 0.95/0.542 | 0.900/0.529 |
| Bitterness/Sweet aftertaste/Teacup preference | 0.882/0.428 | 1.000/0.553 | 1.000/0.528 | 0.947/0.514 | 0.947/0.529 |
| Taste/Bitterness/ Astringency | 0.875/0.400 | 1.000/0.447 | 1.000/0.528 | 0.95/0.542 | 0.9000.529 |
| Taste/Bitterness/Teacup preference | 0.882/0.428 | 1.000/0.500 | 1.000/0.528 | 0.95/0.542 | 0.947/0.529 |
| Bitterness/Astringency/ Teacup preference | 0.882/0.428 | 1.000/0.447 | 0.947/0.500 | 0.904/0.542 | 0.900/0.529 |
| Taste/Sweet aftertaste/Teacup preference | 0.882/0.428 | 1.000/0.553 | 1.000/0.556 | 0.95/0.542 | 0.950/0.558 |

**Table 9.** *Cont.*

| Analysis Item | A Consistency/ Coverage | B Consistency/ Coverage | C Consistency/ Coverage | D Consistency/ Coverage | E Consistency/ Coverage |
|---|---|---|---|---|---|
| | | | Overall Presentation of the Tea (Outcome) | | |
| Taste/Astringency/Sweet aftertaste | 0.866/0.371 | 1.000/0.500 | 1.000/0.551 | 0.95/0.542 | 0.900/0.529 |
| Astringency/Sweet aftertaste/Teacup preference | 0.882/0.428 | 1.000/0.500 | 1.000/0.528 | 0.95/0.542 | 0.947/0.529 |
| Taste/Astringency/Teacup preference | 0.866/0.371 | 1.000/0.500 | 1.000/0.528 | 0.952/0.571 | 0.947/0.529 |
| Bitterness/Sweet aftertaste | 0.882/0.428 | 1.000/0.500 | 1.000/0.556 | 0.947/0.514 | 0.900/0.529 |
| Bitterness/Astringency | 0.882/0.428 | 1.000/0.447 | 0.950/0.527 | 0.904/0.542 | 0.857/0.529 |
| Bitterness/Teacup preference | 0.889/0.457 | 1.000/0.500 | 0.950/0.527 | 0.904/0.542 | 0.900/0.529 |
| Taste/Bitterness | 0.882/0.428 | 1.000/0.500 | 1.000/0.556 | 0.95/0.542 | 0.900/0.529 |
| Taste/Sweet aftertaste | 0.882/0.428 | 1.000/0.553 | 1.000/0.583 | 0.95/0.542 | 0.904/0.558 |
| Astringency/Sweet aftertaste | 0.882/0.428 | 1.000/0.500 | 1.000/0.556 | 0.95/0.542 | 0.900/0.529 |
| Sweet aftertaste/Teacup preference | 0.850/0.485 | 1.000/0.553 | 1.000/0.556 | 0.95/0.542 | 0.950/0.558 |
| Taste/Teacup preference | 0.894/0.485 | 1.000/0.553 | 1.000/0.583 | 0.952/0.571 | 0.950/0.558 |
| Taste/Astringency | 0.882/0.428 | 1.000/0.500 | 1.000/0.556 | 0.952/0.571 | 0.900/0.529 |
| Astringency/Teacup preference | 0.894/0.485 | 1.000/0.500 | 0.950/0.527 | 0.909/0.571 | 0.900/0.529 |
| Taste | 0.894/0.485 | 1.000/0.553 | 1.000/0.583 | 0.909/0.571 | 0.904/0.558 |
| Bitterness | 0.889/0.457 | 1.000/0.500 | 0.952/0.556 | 0.904/0.542 | 0.857/0.529 |
| Astringency | 0.894/0.485 | 1.000/0.500 | 0.952/0.556 | 0.909/0.571 | 0.857/0.529 |
| Sweet aftertaste | 0.850/0.485 | 1.000/0.553 | 1.000/0.583 | 0.95/0.542 | 0.904/0.558 |
| Preference of teacup | 0.863/0.542 | 1.000/0.553 | 0.952/0.556 | 0.909/0.571 | 0.904/0.558 |

However, it is worth noting that the A cup (flaring mouth, wide belly) had the lowest consistency in the causal relationship in Table 9 and the lowest coverage value, while the B cup (contracted mouth, shallow belly) had the most consistent performance in the causal condition combination, with all values at 1. A flaring mouth, wide belly cup has a width of 8.2 cm and height of 3 cm, which is the largest opening among the five cups; the B cup (contracted mouth, shallow belly) has a width of 6 cm and a height of 3.5 cm, indicating the rim thickness of all five cups. The descriptive statistical analysis from Tables 1–6 also showed that when using the A cup (flaring mouth, wide belly), the tea tasted weak, not bitter, not astringent, and without a sweet aftertaste, which meant it was fairly average in terms of overall presentation. When using the B cup (contracted mouth, shallow belly), the tea tasted weak, not bitter, and not astringent, but was fairly average in terms of sweet aftertaste and overall presentation. Therefore, as seen in Fs/QCA, the data could demonstrate the consistency of the causal relationship in a conditional combination that affected the presentation of the tea, but could not reveal the best design for the best overall presentation of the tea. Since the evaluation of Chinese tea differs from the assessment of ordinary beverages, sufficient bitterness and astringency can in fact bring out the sweet aftertaste of the tea. Thus, the descriptive statistical analysis in Tables 2–7 shows that the E cup (open mouth, deep belly) had the highest average in terms of the taste, bitterness, astringency, sweet aftertaste, and teacup preference among the five choices, which demonstrated that E cup's opening diameter of 4.5 cm and height of 3.9 cm (the tallest) offered the best overall presentation for Taiwan's high-mountain Oolong tea. E cup was small and deep enough to preserve the aroma and taste. Therefore, the overall presentation of tea was the best, demonstrating human sensory perception through the "top-down process" and "associative learning" to develop synesthesia of smell and taste [52].

*4.2. Expert Interviews*

After the experiment, interviews were conducted to ask the experts if they knew that the five cups contained the same tea leaves brewed for the same time—in other words, the tea in the five cups was identical. One tea critic indicated that he had wondered about the possibility of using the same tea leaves for testing in this experiment. Two professional teachers of the tea ceremony knew that they were drinking the same tea leaves brewed for the same time, while the other nine subjects were surprised by that fact given that the taste differed in these five cups. One of the nine subjects who was a teacher of the tea ceremony would not believe this fact and insisted the contents were different after sampling them again. One of the tea critics, despite knowing they contained the same tea, maintained that the five cups contained different taste, bitterness, astringency, and sweet aftertaste after carefully sampling them again. After the interviews, the results showed that the A cup (flaring mouth, wide belly) with its wider opening has faster heat and aroma dissipation so yields a less delicious tea, while the E cup (open mouth, deep belly) has a smaller and deeper opening to maintain the temperature and aroma, giving it a stronger taste. Also, its design, taller and slimmer, was preferred by most subjects (nine out of 12). More than half of the experts were satisfied with the presentation of tea in the E cup (open mouth, deep belly). These data could then be cross-analyzed with Fs/QCA data to prove that the teacup shape was a significant factor affecting the taste, bitterness, sweet aftertaste, astringency, and overall presentation of the tea.

Based on the descriptive statistical analysis in Tables 2–7, the highest scores in terms of the taste, bitterness, sweet aftertaste, and teacup preference were counted and the highest average was for the E cup (open mouth, deep belly). Then, the subjects were given the tea in E cup (open mouth, deep belly) and reviewed the taste as being very strong (5 points), very bitter (5 points), very astringent (5 points), and with good sweet aftertaste (5 points), and the ratio of subjects who were very satisfied with the overall presentation (5 points) is shown in Table 10. Three-quarters of the subjects felt that the E cup produced the best-tasting tea, and 50% of subjects believed that the E cup delivered the best sweet aftertaste. Among the five cups, the E cup was the best choice of vessel for Taiwanese Oolong tea, delivering strong taste and bitterness, but also the best sweet aftertaste, and its design was favored by most people. Therefore, the results indicated that the E cup, with its small, deep shape yielding a slim, tall design, was extremely popular. More than half of the experts were satisfied with the presentation of tea in the E cup, which corresponded to Van Doorn et al.'s [23] cross-cultural research on the shape of coffee cups, in which visual information such as the size and height of the coffee cup could affect the consumer's sensory expectations.

**Table 10.** The ratios of subjects' perception of presentation of tea.

| Variables of Tea | Variables of Teacup | F/Total | Pot |
|---|---|---|---|
| Intensity of tea taste | E cup (open mouth, deep belly) | 9/12 | 75 |
| Intensity of tea bitterness | E cup (open mouth, deep belly) | 4/12 | 33.3 |
| Intensity of tea sweet aftertaste | E cup (open mouth, deep belly) | 6/12 | 50 |
| Degree of preference of teacup | E cup (open mouth, deep belly) | 4/12 | 33.33 |

## 5. Conclusions, Limitations, and Future Work

The highest realm of spiritual practice, as stated by the Heart Sutra, is "No forms, sounds, smells, tastes, touchables or objects of Dharma," even though ordinary people are still subject to the sensory inputs from "eyes, ears, nose, tongue, body and consciousness" to yield to "forms, sounds, smells, tastes, touchables and objects of Dharma." Therefore, when savoring tea, an individual can be affected by "eyes, ears, nose, tongue, body and consciousness" to produce the transference of these senses [12]. Through a literature review, experts' experimentation, and expert interviews, as well as a Fs/QCA analysis, the causal relationship between teacup design and the taste, bitterness, sweet aftertaste, astringency, and overall presentation of tea was explored in this study. The study found that the same

tea being placed in cups made of the same material but of different designs would not only affect the taste, but also the bitterness, sweet aftertaste, and astringency, which in turn would influence subjects' perception of the overall presentation of the tea. This proved the relationship between teacup design and the taste of tea recorded in the ancient Chinese books *Tea Shu* and *Treatise on Tea*, and the "associative relations" of teacup design and seasonal tea-drinking habits in the *Chaozhou Tea Classic* [66], in which the teacup shape influences the taste of tea in the process of visual cognition. For designers, it is crucial to have users' insight into tea-drinking vessels to choose a sustainable design. The results generated from the experimentation of the experts (including tea critics, a tea maker, a tea merchant, and professional teachers of the tea ceremony) can provide more accurate and deeper user-centered information for designers.

As Anttila [82] indicated, empirical data in qualitative research can be used as a catalyst in the process of constructing a theoretical discussion. Although this study had a small sample size, China has several thousand years of tea culture and its tea consumption accounts for one-third of the world's tea consumption. Considering the collective environmental impact of consumer behavior, the population of China's tea drinkers play an important role in terms of whether their use of teacups is sustainable. The United National Sustainable Development Goal 12 (Ensure a Sustainable Consumption and Production (SCP) Pattern) defines SCP as "doing more and better with less, increasing net welfare gains from economic activities by reducing resource use, degradation and pollution along the whole life cycle, while increasing quality of life. There also needs to be a significant focus on the supply chain, involving everyone from the producer to the final consumer. This includes educating consumers on sustainable consumption and lifestyles" [38]. While previous studies focused on the visual and material impact of drinking vessels for coffee, wine, and soft drinks, this study considered the small drinking vessels used for the consumption of Chinese tea. Since the tea-drinking population is so large, there is a responsibility to educate consumers about sustainable consumption and lifestyles. The collective power of the sustainable consumption of this great Chinese tea-drinking population should not be ignored. As mentioned in Laakso and Lettenmeier's [84] study on a household-level methodology for the transition towards sustainability, it is believed that the findings of this study can contribute to a more sustainable level of consumption by encouraging relatively few consumption behavior changes in everyday life.

Despite the advancement of science and technology and the improvement of aesthetics, human civilization has had a seemingly endless demand for the use of tea drinking vessels. The perception of objects by human senses, the probability of relying on automated technology, and the complexity of the environment are increasing day by day. From the point of view of satisfying human preferences, the design of tea utensils will always involve some level of materials research and development to meet the needs of tea drinkers. However, due to climate change, the quality of organic farming is much worse than before. If we want to achieve better results from the limited output of the earth, we have to change the design and consumption of drinking vessels. Only in this way can traditional pottery craftsmanship be improved and its role in human life promoted. From the viewpoint of human scientific spirit and materials, the aesthetic and practical design of tea utensils should be both futuristic and sustainable.

The ceramic industry has the same long history and culture as the tea industry. Tea and pottery are inseparable, but as time and technologies advance, teaware is subject to quick elimination due to a loss of practical value, in that many manufacturers or consumers might mistakenly produce or purchase unsuitable teasets. The findings of this study could provide suggestions of better teaware for the ceramics manufacturing industry based on practical design and aesthetics, to deliver perfect-tasting tea, which would not only improve sales, but also reduce the waste of resources involved in producing and eliminating impractical tea utensils, thus furthering the sustainability of the ceramics industry.

There were some limitations to the study. The factors influencing tea-drinking vessels are complex, including the ceramic material, the method of firing, and the chemicals used in the process of glazing the teacups; as exploratory research, this study chose a specific material and color, but future work

could explore the generalizability across different materials and colors. Potential future research might consider the choice of ceramic material and the firing method in terms of the environmental sustainability and tea performance. It is also worth exploring the relationship between the chemical glaze and color of the tea vessels and the quality of the tea itself to incorporate an environmental awareness into the teaware design. While this work highlights expert qualitative research, the qualified participants were few. For future work we recommend adopting a quantitative method to explore a broader base of evidence and different insights for designers and industry. No matter whether one is an educator, a producer, or a consumer, we all need to be conscious of sustainability issues. Therefore, further research on the visual design of teacups for presenting Chinese tea well is recommended because a lot of consumers are not aware that they are using inappropriate or nonfunctional teaware.

**Author Contributions:** S.-C.Y. contributed to the conceptual design of the study, data collection, and drafting the article, and gave final approval. L.-H.P. contributed to the conceptual design of the study and supervision of the progress, and gave final approval. L.-C.H. provided comments, supported the writing of the paper, and reviewed the manuscript.

**Funding:** This research was funded by Sanming University's Scientific Research Start-Up Funds for the Introduction of High-Level Scholars (grant number: 19YG13S).

**Conflicts of Interest:** The authors declare no conflict of interest.

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
