# Peer review of "The Influence of Teacup Shape on the Cognitive Perception of Tea, and the Sustainability Value of the Aesthetic and Practical Design of a Teacup"

_sustainability, doi:10.3390/su11246895_

Round 1

Reviewer 1 Report

When reviewing scientific papers for publication, I usually start with a general overview in terms of a structure, abstract, literature review, methodology, findings of the research, discussion, conclusions, as well as limitations of the study.

To improve the quality of the paper I would suggest to:

1.While the authors stress that the study is subjective, I remain skeptical of the results presented. Whether the study required a control group. Or whether the results would have been different if they had been tested differently.

2.The conclusions of this study on sustainability and innovative design thinking are rather far-fetched and require additional explanation or literature support.

3.Chapters 1 and 2 need to be simplified.

4.Section 2-2, 2-3, 2-4 need to be supplemented with pictures for detailed explanation.

5.The information of subjects in section 4-2 can be written into the experimental design in advance to avoid ambiguity.

6.Supplement the scope and limitations of the study.

Author Response

Comments and Suggestions for Authors

When reviewing scientific papers for publication, I usually start with a general overview in terms of a structure, abstract, literature review, methodology, findings of the research, discussion, conclusions, as well as limitations of the study.

To improve the quality of the paper I would suggest to:

1.While the authors stress that the study is subjective, I remain skeptical of the results presented. Whether the study required a control group. Or whether the results would have been different if they had been tested differently.

Although this study is subjective due to applying the qualitative expert interview,Fs/QCA analysis method is used for quantitative explanation. And this study control the variables on same material and same color of the tea cups to avoid the subjective bias.  Maybe the results have been different, so, at the end, the author suggests future work on quantitative research method.   

2.The conclusions of this study on sustainability and innovative design thinking are rather far-fetched and require additional explanation or literature support.

Have added additional explanation and literature support.

3.Chapters 1 and 2 need to be simplified.

Have been simplified

4.Section 2-2, 2-3, 2-4 need to be supplemented with pictures for detailed explanation.

2-2 to 2-5 are  research results related to the experinmental factors of this study, so I think it’s not necessary to supplement with pictures.  

5.The information of subjects in section 4-2 can be written into the experimental design in advance to avoid ambiguity.

Have been revised and adjusted.

6.Supplement the scope and limitations of the study.

Have supplemented

Reviewer 2 Report

To the authors:

Too much background information and the reader can be distracted. Please summarize the text and focus on the message you want to deliver to your readers Figure 1 has some missing text where only half of the text is shown! The figure has several colors that confuse the reader, and the text is not professionally aligned and formatted. Consider replacing this figure and reproduce it again. Figure 2 is not professionally produced as well. Figures in general are not produced consistently. What are the Chinese symbols in figure 3? All tables need to be professionally presented, and please maintain consistency. The conclusion part should only contain your contribution NOT literature. Please consider moving the literature out of the conclusion part and try to summarize your conclusion clearly.

Author Response

Comments and Suggestions for Authors

To the authors:

Too much background information and the reader can be distracted. Please summarize the text and focus on the message you want to deliver to your readers Figure 1 has some missing text where only half of the text is shown! The figure has several colors that confuse the reader, and the text is not professionally aligned and formatted. Consider replacing this figure and reproduce it again. Figure 2 is not professionally produced as well. Figures in general are not produced consistently. What are the Chinese symbols in figure 3? All tables need to be professionally presented, and please maintain consistency. The conclusion part should only contain your contribution NOT literature. Please consider moving the literature out of the conclusion part and try to summarize your conclusion clearly.

Have been adjusted and revised

Reviewer 3 Report

GENERAL COMMENT

Language is at times quite stilted and a thorough grammatical edit is needed to make sure that the reader understands the intent of the authors.

Some issues are highlighted below.

It is my understanding that the authors propose an investigation into the industrial design of a teacup and if it’s visual aspect of the shape has any impact on the taste of tea; see also line 81.

Also, throughout this paper the term ‘utensil’ is used. Readership might confuse this with a teaspoon. I suggest that the term ‘teacup’ is used throughout.

This research project’s focus is very narrow as only the shape of the teacup is considered. Although the concepts of ‘visual cues’ and ‘sensory stimulation’ are used quite liberally I argue that this includes the consideration of materials, colour and in this case the manufacturing process (e.g firing certain ceramic clay at a higher temperature transforms the clay from earthenware into porcelain.

See for example lines 240-42 “Therefore, teacup functions as a signifier for drinking tea and is visually signified by its design and color, that in the process of visual cognition, the sensation transference of color and shape will occur to produce a specific characteristics of influencing perception of color, aroma and taste of the tea liquid”. Although clearly mentioned in the lit review, the aspects of colour, ceramic material or manufacturing process are not considered and therefore I find the entire project based on very limited information indeed. This puts the believability of the conclusion in doubt. Especially when grand statements are used to justify the outcome, e.g. lines 628-33. Based on this study I doubt if the authors have really contributed to the teacup and tea industries and I doubt ceramists, designers and their respective industries are now able to produce better visual and functional products as investigating into shape alone will not be enough for companies to overhaul their entire teacup production.

LINE

ISSUE

COMMENT

17-22

Explanation of research intent

This part needs rewriting as it is not very clear.

Pls see my comment above as I think that that is the intent of the authors

19-20

This study intended to investigate the visual design of utensil for Chinese tea, in terms of its impact on the taste of the drink by integrating engineers’ and designers’ thinking method for  sustainability of design and industry.

According to the list of ‘experts’ consulted for the investigation (line 556) , no engineers or designers have been consulted. Nor have their comments been clearly cited or their design thinking methods explored in relation to their creative response to the sustainability of teacup design or tea industry

21

integrating engineers’ and designers’ thinking method for sustainability of design and industry.

Sustainability of which design industry? The ceramic industry?

Which other ‘industry’? the tea growing and manufacturing industry? Pls see above and clarify

24

influence on the taste and the scent of tea liquid

influence on the perception of taste and the scent of tea liquid

24-7

a new innovative meaning to the traditional teacup concept design ….

contributing to the sustainable design and development of tea ware industry

This is quite a statement and I haven’t seen any evidence of this?

Are you really contributing to the sustainable design and development of tea ware industry??

37

packed ball tea to the contemporary style of loose tea.

a tightly packed ball tea to the contemporary style of loose-leaf tea.

38

 good utensil and water

 good utensils and water

39

 and utensil is the

 and utensils are the

40

 important tea utensil

 important teacup

41

 medium of visual aesthetics to improve tea quality

Is a teacup improving the quality of the actual tea or the quality of the tea-drinking experience??

42

tea and tea cup of Ming Dynasty

Different tea?

44

with appropriate utensil will

with appropriate utensils will

45-7

What about the teacup being a carrier of ethnic and social culture?

57

quickly replaced and eliminated

quickly replaced or eliminated

58

tea function emphasizes strongly on the aesthetics of tea utensils

I thought the main emphasis on a tea function (ceremony??) is placed on the preparation ritual and the socio-cultural aspects of the tea rather than the aesthetics of the teacup and teapot. This is underscored by your Liu quote

60-2

Therefore, a good cup must be felt by the touch of hand and lips to truly show the warmth and gentleness of combination of tea liquid and the teacup.

This contradicts your earlier statement that the focus is on the aesthetic value of the cup. According to this statement the focus here is on the tactile value of the cup See also previous comment

73-4

The difference between drinking beverage and eating food is the direct contact of oral cavity with the utensil.

?? both cutlery and drinking containers (cups and glasses) you put to your mouth?

So is there really any difference in oral contact?

You are generalizing and therefor your conclusion is false. Regardless of whether there is a difference in oral sensation or not, this is not a reason why people are fascinated to explore the relationship between say a teacup and its perceived improvement of taste.

76-8

found a high correlation of consumer’s perception and behavior toward a beverage with the shape and the color of utensil

What about the material used in the construction of the drinking vessel?

90-3

Not relevant to your investigation as you only focus on the taste sensation impacted by the design of the teacup.

Nowhere in this study are the authors investigating the different ceramic materials or the different firing ranges of the ceramic materials which, surely, do have a major impact on the tea tasting sensation/experience.

This study is purely focused on shape & size. Not colour, materials or ceramic production

131

The Art of Tea-Drinking and the Utensils

No mention of the art of tea drinking/consumption. Only the utensils.

138

Functional Evaluation of Tea

Should this be:

Quality evaluation of tea preparation?

Author Response

Comments and Suggestions for Authors

 GENERAL COMMENT

Language is at times quite stilted and a thorough grammatical edit is needed to make sure that the reader understands the intent of the authors.

Some issues are highlighted below.

It is my understanding that the authors propose an investigation into the industrial design of a teacup and if it’s visual aspect of the shape has any impact on the taste of tea; see also line 81.

Have made revision in the context.

Also, throughout this paper the term ‘utensil’ is used. Readership might confuse this with a teaspoon. I suggest that the term ‘teacup’ is used throughout.

Ok.  But in some parts ‘ drinking utensil’ is still used for better interpretation.

This research project’s focus is very narrow as only the shape of the teacup is considered. Although the concepts of ‘visual cues’ and ‘sensory stimulation’ are used quite liberally I argue that this includes the consideration of materials, colour and in this case the manufacturing process (e.g firing certain ceramic clay at a higher temperature transforms the clay from earthenware into porcelain.

It’s no doubt that there are other factors that can influence the performance of tea liquid, like materials, color and manufacturing process, however, the study limit on the “visual cues” of teacup and the “sensory stimulation”. Therefore, this study control the factors on the same material and same color.

Although this research project focus seems narrow as only consider the shape of teacup, this study used the expert experimental survey design and qualitative interview  to explore the sufficient correlation among variables by Fs/QCA analysis method.

I have made some explanation in the context to avoid the

See for example lines 240-42 Therefore, teacup functions as a signifier for drinking tea and is visually signified by its design and color, that in the process of visual cognition, the sensation transference of color and shape will occur to produce a specific characteristics of influencing perception of color, aroma and taste of the tea liquid. Although clearly mentioned in the lit review, the aspects of colour, ceramic material or manufacturing process are not considered and therefore I find the entire project based on very limited information indeed. This puts the believability of the conclusion in doubt. Especially when grand statements are used to justify the outcome, e.g. lines 628-33. Based on this study I doubt if the authors have really contributed to the teacup and tea industries and I doubt ceramists, designers and their respective industries are now able to produce better visual and functional products as investigating into shape alone will not be enough for companies to overhaul their entire teacup production.

Although the aspects of colour, ceramic material or manufacturing process are the influence factors, those the research field of material engineering. To avoid the intervening factors, this study control the variables on same material and same color of the cups. The author addressed the findings on experts’ experiment and interview for providing a more accurate and deeper insight in user-centered design for designers and producers.  

LINE

ISSUE

COMMENT

17-22

Explanation of research intent

This part needs rewriting as it is not very clear.

Pls see my comment above as I think that that is the intent of the authors

Abstract has been revised.

19-20

This study intended to investigate the visual design of utensil for Chinese tea, in terms of its impact on the taste of the drink by integrating engineers’ and designers’ thinking method for  sustainability of design and industry.

According to the list of ‘experts’ consulted for the investigation (line 556) , no engineers or designers have been consulted. Nor have their comments been clearly cited or their design thinking methods explored in relation to their creative response to the sustainability of teacup design or tea industry

The author intend to explain the different design thinking method between designer and engineer not focus on their design thinking methods. Therefore, it made reader confuse about the intent of this paper. The author has  revised this part.

21

integrating engineers’ and designers’ thinking method for sustainability of design and industry.

Sustainability of which design industry? The ceramic industry?

Which other ‘industry’? the tea growing and manufacturing industry? Pls see above and clarify

The author has  revised this part.

24

influence on the taste and the scent of tea liquid

influence on the perception of taste and the scent of tea liquid

Abstract has been revised.

24-7

a new innovative meaning to the traditional teacup concept design ….

contributing to the sustainable design and development of tea ware industry

This is quite a statement and I haven’t seen any evidence of this?

Are you really contributing to the sustainable design and development of tea ware industry??

Abstract has been revised.

37

packed ball tea to the contemporary style of loose tea.

a tightly packed ball tea to the contemporary style of loose-leaf tea.

Revised.

38

 good utensil and water

 good utensils and water

Revised.

39

 and utensil is the

 and utensils are the

Revised.

40

 important tea utensil

 important teacup

Revised.

41

 medium of visual aesthetics to improve tea quality

Is a teacup improving the quality of the actual tea or the quality of the tea-drinking experience??

Has been revised to “ the quality of the tea-drinking experience”

42

tea and tea cup of Ming Dynasty

Different tea?

Different dynasty has different tea manufacturing process, so the forms of tea leaf are different.

44

with appropriate utensil will

with appropriate utensils will

Revised.

45-7

What about the teacup being a carrier of ethnic and social culture?

Due to the translation mistake, “tea fest” has been revised to “tea ceremony”.In Chinese or Japanese tea way, the teacup arrangement of tea ceremony undertakes various functions.

57

quickly replaced and eliminated

quickly replaced or eliminated

Revised.

58

tea function emphasizes strongly on the aesthetics of tea utensils

I thought the main emphasis on a tea function (ceremony??) is placed on the preparation ritual and the socio-cultural aspects of the tea rather than the aesthetics of the teacup and teapot. This is underscored by your Liu quote

This paragraph is simplified and eliminated.

60-2

Therefore, a good cup must be felt by the touch of hand and lips to truly show the warmth and gentleness of combination of tea liquid and the teacup.

This contradicts your earlier statement that the focus is on the aesthetic value of the cup. According to this statement the focus here is on the tactile value of the cup See also previous comment

This paragraph is simplified and eliminated.

73-4

The difference between drinking beverage and eating food is the direct contact of oral cavity with the utensil.

?? both cutlery and drinking containers (cups and glasses) you put to your mouth?

So is there really any difference in oral contact?

You are generalizing and therefor your conclusion is false. Regardless of whether there is a difference in oral sensation or not, this is not a reason why people are fascinated to explore the relationship between say a teacup and its perceived improvement of taste.

It’s a translation mistake. So it has been modified as following. “The difference between the way of drinking beverage and eating food is the direct contact of lips with the cups or glasses. Therefore, it’s worth investigating   the relationship between the drinking utensil and the taste it brings forth.”

76-8

found a high correlation of consumer’s perception and behavior toward a beverage with the shape and the color of utensil

What about the material used in the construction of the drinking vessel?

Due to the material used in the construction of drinking vessel is belong to the field of material engineer, those studies control the variable on the same material of drinking vessel.

90-3

Not relevant to your investigation as you only focus on the taste sensation impacted by the design of the teacup.

Nowhere in this study are the authors investigating the different ceramic materials or the different firing ranges of the ceramic materials which, surely, do have a major impact on the tea tasting sensation/experience.

This study is purely focused on shape & size. Not colour, materials or ceramic production

Although this study  major on the tea tasting sensation/

experience, the population of tea drinkers do have the the collective influence on environment if their user behavior has been considered into the design of teacups. Line 90-93 have been revised more detail to explain my intent.    

131

The Art of Tea-Drinking and the Utensils

No mention of the art of tea drinking/consumption. Only the utensils.

It should be translated into “The Way of Tea-Drinking and the drinking Utensils”

138

Functional Evaluation of Tea

Should this be:

Quality evaluation of tea preparation?

Has been revised to “Tea sensory Evaluation”

Round 2

Reviewer 1 Report

Overall, the revised version improves the quality of the article.

But I think the relationship between this study and consumption sustainability is still very far-fetched.

As mentioned in this article, the population base of tea drinking in China is huge. But most people often have no research and involvement in tea culture. Therefore, it is debatable for this paper to draw such a macroscopic conclusion based on such subjective feelings of experts.

If the author improves on these problems, it will be an article worth publishing.

Author Response

Comments and Suggestions for Authors

To the authors:

Overall, the revised version improves the quality of the article.

But I think the relationship between this study and consumption sustainability is still very far-fetched.

As mentioned in this article, the population base of tea drinking in China is huge. But most people often have no research and involvement in tea culture. Therefore, it is debatable for this paper to draw such a macroscopic conclusion based on such subjective feelings of experts.

If the author improves on these problems, it will be an article worth publishing.

Dear reviewer,

Thanks for your precious comments and suggestions.  The author has made improvement in the text with yellow marks.  

In this study, considering the complex causality and multiple conditions of the visual perception factors leading to an outcome of the tea liquid performance, the expert experimental subjects were chosen to gather insightful data.  Anttila [78] had pointed out that empirical data in a qualitative research can be used as a basis for description of reality on a complex, holistic, and systemic vision. Ragin & Sedziaka [76] also showed Fs/QCA is best applied when the researcher assumes that complex causality is present and the population of cases is too low for statistical techniques.  Therefore, this study the author opted for an exploratory design with a qualitative approach of expert experimental research.  Although the sample size is too small to represent the big population of China, however, China has several thousand of years of tea culture and its tea consumption accounts for 1/3 of the world’s tea consumption. Hence, the results can be used as catalyst in the process of constructing a theoretical discussion. Therefore, the author considers the collective consumer behavior of environmental impacts. The population of China tea drinkers should be able to play an important role if their consumption behavior of teacups is sustainable.  Based on the study of Laakso & Lettenmeier [80] on household-level methodology for transition towards sustainability, it is believed that the findings of this study have the possibility of achieving a significant and with more sustainable level of consumption by a relatively few consumption behavior change in everyday living. Hence, the author draw a macroscopic possibility, with the collective power of big population in the conclusion of this study.

Reviewer 3 Report

This is a much better paper.

still have one major concern though:

it is clear from the research design that the author's ONLY are concerned with the size and shape of the teacup.

However, lit reviews referenced throughout consider COLOUR as a contributing signifier in addition to shape. See: lines 71 -

it needs to be made VERY clear in the final draft that the authors do not consider colour in this research, only shape.

they will also need to justify the reason why they not consider colour

AND what ceramic product is used in this research project, e.g earthenware? bone china? ironstone china or porcelain?

AND WHY

AND HOW the choice of cup has influenced the perception of the tea

and indentify consideration of colour (and/or ceramic materisl) as a potential future research interest. specifically as the authors try to draw a link between sustainability, aesthetics, visual perception, environmental impact (of chemicals used in the process of firing and glazing/colouring the teacups) etc.

i find it hard to believe that besides shape, colour is not part of an overall 'visual perceptive' experience for any tea drinker.

Based on the comments above i suggest that this paper needs further amendment AND that a 2nd reviewer takes a look

i wish the authors every success with the publication of this paper; i look forward to see it in print. i also look forward to subsequent research into this topic.

sincerely, M. Hulsbosch

Author Response

Comments and Suggestions for Authors

 GENERAL COMMENT

This is a much better paper.

 still have one major concern though:

it is clear from the research design that the author's ONLY are concerned with the size and shape of the teacup.

However, lit reviews referenced throughout consider COLOUR as a contributing signifier in addition to shape. See: lines 71 -

it needs to be made VERY clear in the final draft that the authors do not consider colour in this research, only shape.

they will also need to justify the reason why they not consider colour

AND what ceramic product is used in this research project, e.g earthenware? bone china? ironstone china or porcelain?

AND WHY

AND HOW the choice of cup has influenced the perception of the tea

and indentify consideration of colour (and/or ceramic materisl) as a potential future research interest. specifically as the authors try to draw a link between sustainability, aesthetics, visual perception, environmental impact (of chemicals used in the process of firing and glazing/colouring the teacups) etc.

i find it hard to believe that besides shape, colour is not part of an overall 'visual perceptive' experience for any tea drinker.

Based on the comments above i suggest that this paper needs further amendment AND that a 2nd reviewer takes a look

i wish the authors every success with the publication of this paper; i look forward to see it in print. i also look forward to subsequent research into this topic.

Dear reviewer,

Thanks for your precious comments and suggestions. The author has made improvement in the text with marks.

 Regarding to the variable of colour of the teacup, the author has made statement that due to the different drinking way, observing the tea liquid of Chinese tea is always the major process , in order not to affect the judgement of the color of tea liquid, this study control the color of teacup as white. (see Line246-251 & Line324-326)

In the conclusion, the author has made some suggestion and direction on ceramic material, the way of firing ceramic, and the chemicals used in the process of glazing the teacups for potential future research.  

Round 3

Reviewer 1 Report

After several rounds of revision, I think the quality of the article has been improved and can be published

Author Response

Dear reviewer,

Thanks for your precious comments and suggestions.  The author has made English and styles editing in the attached text with marks.

Best Regards,
